# Integrated MRI–Immune–Genomic Features Enclose a Risk Stratification Model in Patients Affected by Glioblastoma

**DOI:** 10.3390/cancers14133249

**Published:** 2022-07-01

**Authors:** Giulia Mazzaschi, Alessandro Olivari, Antonio Pavarani, Costanza Anna Maria Lagrasta, Caterina Frati, Denise Madeddu, Bruno Lorusso, Silvia Dallasta, Chiara Tommasi, Antonino Musolino, Marcello Tiseo, Maria Michiara, Federico Quaini, Pellegrino Crafa

**Affiliations:** 1Medical Oncology Unit, University Hospital of Parma, 43126 Parma, Italy; giulia.mazzaschi@unipr.it (G.M.); alessandro.olivari@unipr.it (A.O.); chiara.tommasi@unipr.it (C.T.); antonino.musolino@unipr.it (A.M.); marcello.tiseo@unipr.it (M.T.); michiara@ao.pr.it (M.M.); 2Department of Medicine and Surgery, University of Parma, 43126 Parma, Italy; pellegrino.crafa@unipr.it; 3Neuroradiology Unit, University Hospital of Parma, 43126 Parma, Italy; apavarani@ao.pr.it; 4Pathology Unit, University Hospital of Parma, 43126 Parma, Italy; costanzaannamaria.lagrasta@unipr.it (C.A.M.L.); caterina.frati@unipr.it (C.F.); denise.madeddu@unipr.it (D.M.); brunolorusso@gmail.com (B.L.); 5Geriatric Department, University of Genoa, 16132 Genova, Italy; silvia.dallasta8@gmail.com

**Keywords:** glioblastoma, tumor immune microenvironment (TIME), magnetic resonance imaging (MRI)

## Abstract

**Simple Summary:**

Despite crucial scientific advances, Glioblastoma (GB) remains a fatal disease with limited therapeutic options and a lack of suitable biomarkers. The unveiled competence of the brain immune system together with the breakthrough advent of immunotherapy has shifted the present translational research on GB towards an immune-focused perspective. Several clinical trials targeting the immunosuppressive GB background are ongoing. So far, results are inconclusive, underpinning our partial understanding of the complex cancer-immune interplay in brain tumors. High throughput Magnetic Resonance (MR) imaging has shown the potential to decipher GB heterogeneity, including pathologic and genomic clues. However, whether distinct GB immune contextures can be deciphered at an imaging scale is still elusive, leaving unattained the non-invasive achievement of prognostic and predictive biomarkers. Along these lines, we integrated genetic, immunopathologic and imaging features in a series of GB patients. Our results suggest that multiparametric approaches might offer new efficient risk stratification models, opening the possibility to intercept the critical events implicated in the dismal prognosis of GB.

**Abstract:**

Background: The aim of the present study was to dissect the clinical outcome of GB patients through the integration of molecular, immunophenotypic and MR imaging features. Methods: We enrolled 57 histologically proven and molecularly tested GB patients (5.3% IDH-1 mutant). Two-Dimensional Free ROI on the Biggest Enhancing Tumoral Diameter (TDFRBETD) acquired by MRI sequences were used to perform a manual evaluation of multiple quantitative variables, among which we selected: SD Fluid Attenuated Inversion Recovery (FLAIR), SD and mean Apparent Diffusion Coefficient (ADC). Characterization of the Tumor Immune Microenvironment (TIME) involved the immunohistochemical analysis of PD-L1, and number and distribution of CD3+, CD4+, CD8+ Tumor Infiltrating Lymphocytes (TILs) and CD163+ Tumor Associated Macrophages (TAMs), focusing on immune-vascular localization. Genetic, MR imaging and TIME descriptors were correlated with overall survival (OS). Results: MGMT methylation was associated with a significantly prolonged OS (median OS = 20 months), while no impact of p53 and EGFR status was apparent. GB cases with high mean ADC at MRI, indicative of low cellularity and soft consistency, exhibited increased OS (median OS = 24 months). PD-L1 and the overall number of TILs and CD163+TAMs had a marginal impact on patient outcome. Conversely, the density of vascular-associated (V) CD4+ lymphocytes emerged as the most significant prognostic factor (median OS = 23 months in V-CD4^high^ vs. 13 months in V-CD4^low^, *p* = 0.015). High V-CD4+TILs also characterized TIME of MGMT^meth^ GB, while p53^mut^ appeared to condition a desert immune background. When individual genetic (MGMT^unmeth^), MR imaging (mean ADC^low^) and TIME (V-CD4+TILs^low^) negative predictors were combined, median OS was 21 months (95% CI, 0–47.37) in patients displaying 0–1 risk factor and 13 months (95% CI 7.22–19.22) in the presence of 2–3 risk factors (*p* = 0.010, HR = 3.39, 95% CI 1.26–9.09). Conclusion: Interlacing MRI–immune–genetic features may provide highly significant risk-stratification models in GB patients.

## 1. Introduction

The growth characteristics and spread dynamic coupled with treatment resistance make glioblastoma (GB) among the deadliest of all cancers, with a median survival rate of 14–16 months despite intensive regimens including surgery, radiotherapy, and chemotherapy [1,2]. 

The imperative need for prognostic and predictive markers of this lethal disease has substantially prompted the clinical and molecular characterization of GB, leading to the discovery of key genetic alterations, included in the last updated WHO classification, which are now guiding the current therapeutic strategies of brain tumors [3,4]. The identification of IDH-1 gene mutation endowed with a favorable prognostic value [5,6], and the observation that MGMT-gene methylation confers sensitivity to temozolomide-based chemotherapy have represented a breakthrough in the clinical management of GB [7,8,9]. Other gene alterations involving EGFR [10], p53 [11], ATRX and PTEN [12] have added new insights in the pathobiology of GB, although their significance and translation in a clinical setting has not been fully established.

To pursue the path towards novel reliable prognostic markers in GB, focus has recently been shifted to immune-centered cues, partly as a reflection of the bursting advent of Immune Checkpoint Inhibitors (ICIs) in the therapeutic scenario of solid and hematologic malignancies. The considerable number of reported and ongoing clinical studies on GB immunotherapy [13] has been promoted by advanced knowledge on the anatomical and functional aspects of the brain immune system. Specifically, the claimed documentation of brain lymphatics [14,15] and the notion that, in addition to macrophages and dendritic cells, resident populations of microglial cells display antigen presenting cell (APC) properties [16], strongly support the GB immunity cycle as a potential therapeutic target. Attempts to treat the disease with ICIs have substantially failed to achieve a durable response, highlighting the long-lasting knowledge on the distinctive characteristics of the brain immune contexture and its neoplastic counterpart [17,18]. While the Tumor Immune Microenvironment (TIME) in GB has been generally ascribed as immunosuppressive [16], recent findings have called into question whether the prominent incidence of Tumor Associated Macrophages (TAMs) exclusively exerts a negative role or whether Tumor Infiltrating Lymphocytes (TILs) consistently contribute to a clinically favorable TIME [19,20]. In addition, as the complex GB cytoarchitecture is a critical factor for tumor progression and patient outcome, the cross-talk between immune cells and vascular structures is under intense scrutiny [21]. Thus, more reliable prognostic factors may be obtained by the assessment of the relative proportion and topographical distribution of different immune phenotypes within the TIME [22]. 

Great efforts have been also addressed to GB imaging techniques able to provide clinically meaningful elements through the extraction of defined parameters from Magnetic Resonance Imaging (MRI) [23,24,25]. First, morphologic imaging can provide several details regarding tumor heterogeneity, hypercellularity, blood–brain barrier disruption, necrosis, hemorrhagic foci, mono or multifocal lesions and mismatch FLAIR/T2. Moreover, quantitative multimodal imaging with perfusion techniques and spectroscopy can add important aspects such as tumor margins invasion, vasculature patterns, permeability and evaluation of metabolites within the lesion (i.e., 2-Hydroxyglutarate related to IDH1 mutation) through radiomic studies [23,26]. Furthermore, over the last few years, there has been growing interest towards artificial intelligence to images microstructure analysis and the extraction of quantitative variables potentially linked to biomolecular and prognostic cues [25,27,28]. For example, mean apparent diffusion coefficient (mean ADC), which directly correlates to the more solid or liquid consistency of the tumor, has been documented as a potential predictor of tumor grade and patient prognosis [29,30]. Conversely, limited investigations are available on the potential association of radiomic with TIME parameters, leaving underexplored the possibility to non-invasively decode the key immune characteristics underpinning survival outcome and therapy resistance in GB patients [31,32]. 

Along these lines of investigation, we carried out a multiparametric analysis on 57 GB cases to determine whether clinico-pathological, biomolecular and MR imaging features reflect specific immune contextures, with the ultimate aim to offer prognostically relevant radio-immune signatures.

## 2. Materials and Methods

### 2.1. Patient Population

Data were retrospectively analyzed on patients diagnosed and surgically treated for Glioblastoma at the University Hospital of Parma (Medical Oncology, Neuroradiology and Neurosurgery Units) between January 2012 and January 2018. The clinico-pathological characteristics of our patient population are summarized in Table 1, while the clinical course of each patient is illustrated by the swimmer plot (Figure 1). 

Most patients were treated with standard procedures mainly consisting of surgery, STUPP protocol including chemotherapy (CHT) plus radiotherapy (RT) and maintenance CHT with Temozolomide. The vast majority of patients were from the Oncology Unit of University Hospital of Parma, while a few were from Piacenza and Reggio Emilia Hospitals.

GB patients who fulfilled the following criteria were included in the present study:Availability of tissue specimens as a result of surgical resection or biopsy;Histopathological diagnosis of glioblastoma made by an expert pathologist;Molecular and immunophenotypic characterization of pathologic specimens;Data derived from MRI pre-surgical images (either on 1.5T or 3T);Availability of complete clinical records;Set date of last follow-up if the patient(s) was still alive on 30 September 2021.

Patients were enrolled after informed consent and the study was performed following the approval from the ethical committee (1116/2019/OSS*/AOUPR) and in accordance with Helsinki principles.

### 2.2. Genetic-Molecular Analysis (MGMT, IDH1, p53, EGFR, ATRX)

MGMT methylation: biopsy specimens were treated and analyzed by an expert pathologist (P.C.) at the University-Hospital of Parma. MGMT methylation status was evaluated by end-point amplification of the extracted DNA and genotypization and allelic quantification of the sequence of interest. A consistent group of patients (*n* = 28) were evaluated with methylation specific PCR (MSP), as the study leads back to a time before the adoption of routine pyrosequencing analysis (PSQ) of MGMT. PSQ was then adopted for MGMT analysis of the remaining 29 cases using PyroMark Q96 ID Instrument (Qiagen SRL, Milan, Italy). General guidelines published in the literature [33] were adopted to interpret the results and samples were considered methylated if >9%.

IDH, ATRX, EGFR and P53 were detected by immunohistochemistry (IHC) according to standard procedures employed for diagnostic pathology (OptiView DAB IHC Detection Kit—Ventana Medical Systems). Briefly, sections from formalin- fixed-paraffin-embedded (FFPE) samples were treated with 3% H2O2 for 5 min at room temperature (RT) to block endogenous peroxidase. For antigen unmasking, slides were heated in sodium citrate buffer (10 mM sodium citrate, pH 6.0) for 15 min at 100 °C. After washing with phosphate-buffered saline, sections were immunostained with anti-human IDH1-R132H antibody (H09, Ventana) or anti-human ATRX antibody (ab97508, Abcam), and incubated at 4 °C over night. CONFIRM anti-EGFR (5B7, Ventana) rabbit monoclonal antibody was used to quantitatively detect the presence of EGFR overexpression within the specimens. Anti-p53 primary antibody (Bp53.11, Ventana) was used to detect the overexpression of the phosphoprotein. Ultraview universal DAB detection kit was applied to reveal immunoreactions. 

Negative controls consisted of samples subjected to the same immunohistochemical procedure in which the primary antibody was omitted.

Data were qualitatively and quantitatively revised by an expert pathologist (P.C.).

### 2.3. Immunohistochemical Analysis of TIME

An extensive characterization of TIME was carried out on 32 GB samples and involved the analysis of Tumor Infiltrating Lymphocytes (TILs), Tumor Associated Macrophages (TAMs) and PD-L1 levels of expression.

Tissue sections (5 μm thick) from FFPE blocks containing representative tumors were processed for IHC. 

The density and distribution of immune cells and the expression of PD-L1 (Appendix A) were evaluated using a computerized image analysis system (Nanozoomer-Digital-Pathology apparatus, Hamamatsu Photonics). Photomicrographs covering the entire neoplastic area were digitally obtained and recorded for later analysis, allowing large-scale histological evaluation with high precision across each sample. Necrotic and hemorrhagic areas were excluded from counting to avoid potential artifacts due to unspecific staining. Distinctive and consistent areas displaying clear tumor margins could be detected in only 30% of cases, preventing a suitable assessment of TILs incidence at the cancer/brain interface.

-TILs were analyzed by the immunohistochemical detection of CD3, CD8 and CD4. Immunoperoxidase was performed by an automated staining system (OptiView DAB IHC Detection Kit—Ventana Medical Systems) using antibodies against CD3 (clone 2GV6), CD8 (clone SP16) and CD4 (clone SP35). We adopted the basic morphometric principle by counting the number of positive cells in the microscopic fields of the defined area. Only small, mostly round shaped, nucleated cells with intense surface immunoperoxidase labelling were considered, while cells with elongated cytoplasm or faint staining were excluded. Thus, the density (*n*/mm^2^) of CD3+, CD8+, and CD4+ cells was computed analyzing a tissue area of a minimum of 6.83 mm^2^ to a maximum of 254.99 mm^2^ according to the size and quality of samples. The incidence of TILs phenotypes was evaluated according to their localization in direct contact with neoplastic cells (intratumoral, IT) or in perivascular (PV) or intravascular (IV) location. TILs localized within 20 μm linear distance from vascular profiles were defined as PV while IV when lymphocytes were endowed within the vessel wall. The cut off distance of 20 µm was selected according to the conventional view that it represents a minimal distance allowing a bio-humoral cross talk. Vascular profiles were morphologically identified, however, to better define their interaction with TILs, sections were stained automatically (OptiView DAB IHC Detection Kit—Ventana Medical Systems) with anti-CD31 (mouse, Ventana) and -CD34 (mouse, Ventana) antibodies or manually with α-smooth muscle actin (SMA) (mouse, Abcam, 1.5 h at 37 °C) antibodies. In addition, to ascertain TILs localization with respect to vascular profiles, CD4 and α-SMA were simultaneously detected by double immunofluorescence in a subset of samples. To this purpose, following incubation for 1 h at 37 °C with respective primary antibodies, FITC- and TRITC-conjugated secondary antibodies were applied for 1 h at 37 °C and nuclei were visualized following 20 min exposure to 4′,6-Diamidino-2-phenylindole (DAPI; D8417, Merck, NJ, USA). Examples of the immunohistochemical detection of each investigated phenotype and its spatial distribution within the tissue are provided in Appendix A.-TAMs were detected by immunoperoxidase through an antibody directed against CD163 (clone MRQ-26). Due to the irregular and wavy profile of macrophages hampering the precise definition of individual cells, we measured the fractional area occupied by CD163 immunolabeling using a software dedicated to image analysis (Image Pro Plus 4.0, Media Cybernetics, Rockville, MD 20852, USA).-PD-L1 was assessed by a specific antibody (clone SP263) and quantified using an algorithm to obtain the tumor PD-L1 score (H-score; 0–300) on the basis of both extent and intensity of PD-L1 staining [34]. PD-L1 expression was also measured in stromal compartments by a semi-quantitative approach using a grading score from 0 to up to 3+ according to staining intensity.-Controls for each investigated antigen were represented by sections undergoing the same staining protocol but omitting the primary antibody or using an indifferent antibody.

### 2.4. MRI-Based Texture Analysis

The entire population of 57 patients enrolled in the study had a brain MRI scan before surgery. The study protocol consisted of the acquisition of the same sequences between studies: T1 3D before and after the administration of contrast medium, Axial FLAIR 4 mm, Axial GRE 4 mm sequence, Axial T2 4 mm, Axial DWI with EPI technique with B values between 0–1000 Gs. Computation of the ADC map with quantitative values in mm^2^/s was also performed. Two MRI scanners were used: a 3.0 T system in 36 patients (63%) and 1.5T System in 21 patients. Using a GE software of synchronization (Ready-View), an expert Neuroradiologist manually outlined a bidimensional free ROI on the widest tumoral enhancing axial diameter on T1 post-contrast sequence and then cloned it on the other three sequences (FLAIR, ADC, GRE). From the bidimensional ROI were extracted the following parameters: maximum tumor area (mm^2^), SD FLAIR, mean ADC (mm^2^/s) and SD ADC (mm^2^/s) values. The rationale behind the MRI characterization is that SD ADC and FLAIR are directly related to tissue heterogeneity within the tumor, as the SD value itself is a measurement of the dispersion of values around the mean, and statistically correlates with the homogeneity or heterogeneity of the set itself [35]. Moreover, mean ADC correlates with the mobility of water molecules within a lesion, which is linked to the structural features (being either more “solid” or “soft”) of the mass itself [35]. 

### 2.5. Statistical Analysis

OS and PFS were estimated according to the Kaplan–Meier method. OS was defined as the interval from the date of surgery to the date of death or to the date of last follow-up for alive patients. PFS was defined as the interval from the date of surgery to the date of radiological/clinical progression or death due to any cause, whichever occurred first, or to the date or last follow-up visit for patients alive without disease progression. 

Cut-off for survival analysis was set at 30 September 2021. Median follow-up was calculated according to the so-termed “reverse Kaplan–Meier” (Kaplan–Meier estimate of potential follow-up) technique [36]. Log-rank test (Mantel-Cox) was applied to evaluate statistical differences in OS and PFS between groups. Survival data were then analyzed through Cox uni- and multivariate proportional hazards regression models and results expressed as hazard ratios (HR), 95% CI and *p* values. The multivariate models were fitted, including the covariates which were statistically significant in the univariate model.

The Fisher’s exact test was used to examine the differences between categorical variables while the Mann–Whitney U test or Kruskal–Wallis to detect differences in continuous variables between groups of patients, given that the distribution of data was not normal (Kolmogorov– Smirnov test). Classification and regression tree (CART) analysis identified specific cut-off values that segregated patients by clinical outcomes.

*p* value of 0.05 was set as a threshold of statistical significance. To minimize the risk of multiplicity, the Bonferroni correction test was applied to all our multiple comparisons.

IBM SPSS Statistics v 25.0 (IBM) and Stata 13 with Cart module (Statacorp) were used to perform all computational analyses.

## 3. Results

### 3.1. Patients Characteristics

The study population included 57 subjects affected by glioblastoma and admitted to our Institution from January 2012 through January 2018. Median age was 63 years (range: 41–82), with a slight male prevalence (57.9%) (Table 1).

At MRI, the vast majority of cases presented a single tumor lesion (78.9%) and most primary tumor sites were frontal (45.6%) and temporal (35.1%), while parietal (8.8%), occipital (7%), cerebellar (1.8%) and deep (1.8%) lobes were significantly less represented.

With a median follow up of 51.5 months (95% CI 29.9–73.2), median OS of our patient population was 16.1 months, ranging from 4.6 to 87 months, while median PFS resulted in 6.3 months (95% CI 3.2–9.4).

### 3.2. Correlations between Genetic, MRI and TIME Characteristics

Most GB cases resulted in IDH1-2 wt (87.7%) while p53 mutation was detected in more than 60% of cases (Table 1).

As reported in Table 2, mean ADC values ranged from 0.657 to 3.430 × 10^−3^ mm^2^/s, while SD ADC values from 0.32 to 9.15 × 10^−4^ mm^2^/s (median: 2.95 × 10^−4^ mm^2^/s) and SD FLAIR from 0.12 to 7.38 × 10^−4^ (median: 1.12 × 10^−4^). Furthermore, we found a significant relationship between SD ADC and IDH1 mutation, since higher SD ADC values were documented in IDH1 mutant cases (*p* = 0.028). No significant associations between MGMT methylation status and MRI findings were detected.

The tissue density (*n*/mm^2^) of T-cell populations, as assessed by IHC, largely reflected the expected incidence according to their phenotype, with a higher representation of CD3+ T cells over a similar lower fraction of CD4+ and CD8+ lymphocytes. When TILs were quantified according to the different location within TIME, we observed that 74.5% of the overall number of CD3+ T cells were in contact with GB cells (IT), while the remaining were associated with vascular structures.

Accordingly, 19.4% and 18.7% of the overall number of CD4+ and CD8+ lymphocytes, respectively, were located at vascular sites where a higher CD4-to-CD8 ratio (1.15) compared to that observed at tumor site (IT, 0.89) was detected (Table 3; Figure 1).

TILs: Tumor Infiltrating Lymphocytes; IT: intratumor; PV: perivascular; IV: intravascular.

Macrophages were evaluated here by the expression of CD163, a M2 associated antigen which is considered the most suitable makers of TAMs [37,38] and known to be involved in GB oncogenesis [39]. Although CD163+ cells were observed in perivascular space (Figure 1) and in inflammatory areas of potential glial polarization, we focused our quantification on the tumor core. The fractional area occupied by TAMs significantly varied among cases reaching more than 2% of the entire tissue in 32% of GB samples.

Strong tumor PD-L1 expression (H-Score > 150) (Figure 2, Appendix A) was present in only 15% of cases and intermediate (H-Score: 30–149) in 12% while negative staining was predominant (73%). GB cells displayed membranous and cytoplasmic expression as specific PD-L1 labelling was noted on spared neural cells (Appendix A). Amplification of PD-L1 signal in tumor cells located in proximity to vascular structures was noted (Figure 2, Appendix A) and stromal PD-L1 expression was also apparent in 14 out of 34 examined tumors.

Studying the impact of the genetic-molecular background on TIME and MRI features, we intriguingly found that cases carrying IDH1-2 mutation, although representing only 5% of the overall population, were characterized by a significantly higher density of PV-TILs (CD3+, CD4+ and CD8+, *p* < 0.005) compared to WT ones (Figure 3A).

A trend towards TIME particularly enriched in perivascular CD3+ (*p* < 0.05), CD4+ and CD8+ (NS) was distinctive of MGMT methylated cases (Figure 3B), which also frequently presented with only one brain lesion at diagnosis (*p* < 0.05). Interestingly, when we categorized patients in long- (OS ≥ 24 months) versus short-term (OS < 9 months) survivors, we observed a higher proportion of MGMT methylated cases (82% vs. 47% in the short-term, *p* = 0.07) coupled with significantly higher MR-derived Mean ADC (16 × 10^−4^ vs. 9 × 10^−4^ mm^2^/s) and vascular CD4+ TILs (95.2 vs. 8.4 *n*/mm^2^, *p* = 0.05) (data not shown).

Among other genetic-molecular characteristics, EGFR overexpression appeared to condition increased PD-L1 levels of expression (*p* < 0.05, Figure 3C), while p53 mutation did not display a substantial impact on tissue or imaging parameters (not shown).

When we focused on the differential distribution of TIME features according to MR-based imaging, no significant correlations were observed (Spearman test—data not shown).

### 3.3. Impact of Genetic, MRI and TIME Features on Survival Outcome

Our study confirmed the well-known correlation between MGMT methylation and clinical outcome. Median OS in highly methylated patients was 20.5 months (10.6–30.4 CI) vs. 14.8 months (12.8–16.8 CI) in not methylated (*p* = 0.007) (Figure 4A). Moreover, although within a limited sample size, IDH1-wt patients had a significantly shorter median OS compared to IDH mutated cases (15.03 months vs. NR, *p* = 0.042). Conversely, EGFR and p53 mutation did not appear to affect survival outcome (Table 4).

No relevant impact on OS was documented by MRI-derived SD ADC, SD FLAIR and Max area of Tumor enhancement (Table 4). However, when we applied CART Tree Regression Analysis to identify specific cut-offs for mean ADC (1.48 × 10^−3^ mm^2^/s), two subgroups of high (*n* = 44) and low (*n* = 13) patients with distinct clinical outcomes were defined. Mean ADC^high^ cases had a significantly (*p* = 0.007) prolonged OS (median OS: 24.01 months, 95% CI 8.85–39.17) compared to mean ADC^low^ group (15.03 months, 95% CI 12.7–17.36), thus implying the relevant prognostic power of this MRI feature (Figure 4B).

Among TIME parameters, the extent of CD163+ TAMs, PD-L1+ levels and the overall number of CD3+, CD4+ and CD8+ TILs were unable to strongly discriminate patients’ survival. Conversely, the incidence of CD4+ lymphocytes associated with vascular structures had significant prognostic power. Specifically, GB patients with high PV-CD4+ cells had a median survival of 23.6 months (95% CI 12.8–34.3) compared to 14.28 months in the low PV-CD4+ group (Figure 5A). Similarly, median OS in cases displaying a high number of IV-CD4+ TILs resulted in a significant increase (*p* = 0.008) compared to that of patients with low IV-CD4+ (20.49 vs. 11.71 months) (Figure 5B). When we combined PV and IV values to obtain the overall incidence of vascular-associated (V) CD4+ cells, prolonged OS was observed in GB patients carrying high V-CD4+ lymphocytes (Figure 5C).

Integrating the most prognostically relevant variables (MGMT, mean ADC and V- CD4+ TILs), we developed a risk stratification score. One point was assigned to each low V-CD4+ TILs, MGMT not methylated and low mean ADC values as negative predictors. By applying this multiparametric strategy, GB patients with a risk score of 0–1 had significantly (*p* = 0.010) prolonged OS (median OS 20.49 months, 95% CI, 0.0–47.37) compared to those with scores 2–3 (median OS 13.22 months 95% CI, 7.22–19.22) (Figure 6).

Finally, in view of the well-known notion that IDH mutant status conditions a significantly better prognosis, we performed additional survival analyses considering IDH1-2 WT cases only (*n* = 53) to strengthen our findings independently from IDH status. As reported in Appendix A, the impact of our previously identified prognostic factors (MGMT status, MR-based Mean ADC, CD4+ vascular TILs and integrated risk score) on OS was confirmed regardless IDH status.

## 4. Discussion

Despite important advances in genetic characterization and unveiled properties of the tumor immune background, therapy resistance remains an unsolved issue in the clinical management of GB. The difficult accessibility and often inadequacy of tumor biopsies further limit the possibility of defining, monitoring and positively impacting on critical events implicated in the evolution of this devastating disease. Challenging efforts are currently being undertaken to intercept the remarkable heterogeneity and the unique immune-vascular interplay of brain tumors. Multidisciplinary studies aimed at providing insights to the composition of TIME potentially deciphered by high throughput extracted MR images might implement our approach to GB. Several observations are also emerging on the attempt to translate through advanced MRI imaging relevant biomolecular tumor hints. Thus, based on the integration of genetic, imaging and tissue immune-vascular features, results of the present study revealed specific profiles sharply dissecting GB clinical outcome.

The genetic background of our limited cohort of patients largely reflects the incidence of alterations, as MGMT and IDH1, commonly tested in GB. While the well-established prognostic relevance of MGMT methylation status [40,41,42] was not associated here with specific imaging parameters, IDH1 mutation was correlated with SD ADC. This evidence has not been reported in the literature, while other studies [43,44,45,46] have documented an association of IDH1 mutation with mean ADC, being low values predictive of IDH1-wt status. As mean ADC correlates with tumor cellularity and tissue consistency, its potential implication as a prognostic parameter has also been well described [46,47,48]. This finding was largely confirmed here, making it reasonable to infer that GB displaying low mean ADC, underlying a more solid consistency and high cellularity, are characterized by a more aggressive behavior and rapid spread. Thus, mean ADC and IDH1 status appear to be closely linked in conditioning GB patients’ survival and their prognostic role should be tested in a larger and more representative cohort of IDH1-mutant cases.

Interrogating TIME to uncover the potential link with the genomic and clinical characteristics of our patient population, we found that high V-CD4+ lymphocytes content was a distinctive feature of IDH1-mutant GB, and when combined with mean ADC, ultimately portrayed an MRI–immune–genetic trait. MGMT methylation also trended toward a TILs-rich microenvironment, however without significant association with MRI parameters.

It should be pointed out that, while several reports have documented a correlation between IDH status and tumor infiltrating lymphocytes [17,49], the literature is scant about the identification of specific immunophenotypic characteristics associated with MGMT status and so far inconsistent results have been described [50,51]. On a large-scale RNAseq profiling of 769 GBM patients from five independent datasets, the score evaluation, defined as GBM-associated TIME immune cell infiltration (GTMEI) score, of more than 20 immune cell marker genes, including Thy-1 CD4, were analyzed in patients with different MGMT methylation status to define prognostic classes [52] and/or response to treatment [53]. While immune gene signatures were able to discriminate high- vs low-risk groups and to predict the response to chemo- or immuno-therapy, no differences were observed in the GTMEI score according to MGMT methylation status. Additionally, these findings are in line with data from clinical trials, showing that sensitivity to immune checkpoint inhibitors is not affected by MGMT status [54].

Epigenetic mechanisms triggered by cancer to shape the identity of tumor infiltrating CD4+ T cells within the TIME have been observed in cancer models [55,56]. Moreover, evidence has been provided on the possibility that GBM, through DNA methylation of key genes, dictates the fate of tumor infiltrating CD4+ T cells [57]. Unfortunately, in this small patient cohort only, one case had documented, MGMT promoter methylation, hampering a conclusive view on the potential involvement of MGMT status in conditioning the extent and function of CD4+TILs in GBM, as suggested by our findings.

The ability of MRI features to reflect TIME and the pathologic clues of GB remains an unsolved issue, although the potential to non-invasively predict patient outcome and response to treatment is of paramount relevance. The GB immune landscape has been extensively investigated, as properly reported by Quail et al. [16], and more recently revisited by RNAseq and FACS single cell profiling in a comparative analysis with brain metastasis [58,59]. In addition, to highlight the inductive role of cancer lineage on TIME composition, these observations have called into question the validity to apply to GB the common criteria employed in solid tumors to assess the prognostic and predictive value of tissue immune descriptors. Even the typical ascription to classical phenotypes, such as M1/M2 macrophages, cannot be ascertained in the context of GB [58,59,60]. These contentions, together with the limited clinical significance of Tumor Mutational Burden (TMB), have further tangled the road to define valid biomarkers in a GB setting. In this regard, actual interest has been shifted toward the intense immune-vascular interplay taking place in GB in reason of novel methodological approaches [61], and biologic discoveries on tumor immunity [62] and angiogenesis [63].

The development of strong immune reactions against tumor antigens is accomplished by their entry into fluid spaces and egress into draining lymph nodes or, when present, into local so called Tertiary Lymphoid Structures (TLS). This widely accepted principle is not easily applicable to or detectable in GB due to the peculiar brain vasculature, including lymphatic vessels, and the substantial lack of lymphoid tissue limiting viable immune synapses. The fact that microglial cells might function as APC [20] could represent a vicariant mechanism allowing the local initiation of the GB immunity cycle. The profound vascular rearrangement characterizing GB involves a variety of cellular and biological processes, including the existence of tumor CD133+ multipotent cells capable of endothelial lineage differentiation [64] and the repeatedly documented dysregulation of pro-angiogenic and anti-angiogenic pathways. The mutual role played by vascular remodeling and immune activation in dictating the response to immunotherapy in GB has been recently emphasized by the observation that CD4+ lymphocytes are primary actors of the so-called tumor vessel normalization [65]. This process mainly consists of restoration of pericyte coverage, improved vessel perfusion/permeability, ultimately leading to attenuated hypoxia [66,67]. Importantly, disruption of vascular normalization negatively affects immune activation while conditional CD4+ lymphocytes knockout alters vascular remodelling [68]. Taken together, these observations underscore the dual function of type 1 T helper (T_H_1) CD4+ cells in shaping both vascular and immune tumor compartments, strongly suggesting their determinant role in the outcome of immune checkpoint and angiogenetic targeting.

Along these lines, findings of the present investigation on the prognostic impact of vessels associated TILs strongly support the reported relevance of CD4+ lymphocytes in the immune-vascular crosstalk critically implicated in cancer growth and regression.

Finally, it is increasingly clear that the heterogeneous nature of GB imposes a dramatic improvement in our actual knowledge, likely gained from genetic and immune profiling, in order to develop future targeted therapeutic strategies. Great efforts have been addressed to molecular underpinnings of GB, by high-throughput single omic profiling (i.e., whole genome sequencing, RNA sequencing, deep metabolomics) [69] with the aim to define new contexts of vulnerability. Nonetheless, a real multi-omic approach should not be restrained to genetic features only [69,70], but potentially extended towards multiple aspects, including imaging and immunologic cues. From this perspective, our study might represent a preliminary attempt to merge tumor- and patient-specific characteristics, achievable by different areas of expertise, thus more faithfully intercepting disease heterogeneity and its therapeutic implications.

Despite its novelty, several limitations of the present study must be acknowledged. The observational and retrospective nature of the present study, together with the relatively limited sample size, does not allow an immediate translation of our proposed approach into clinical practice. In addition, a more detailed immunophenotypic characterization of TILs, and, to a further extent, of CD4+ subpopulations (i.e., Tregs, Th17, Ki67, PD-1), should be performed to implement the scientific significance of our results. The role of other immune relevant phenotypes, such as Myeloid-Derived Suppressor Cells, was not addressed here as they may well participate to condition the immunosuppressive trait of TIME in GB. Finally, it is worth mentioning that our study was performed on patients enrolled from 2012 to 2018, before the introduction of the revised 2021 WHO classification of brain tumors. Thus, the three cases carrying the IDH mutation should be actually diagnosed as astrocytoma G4. We will make an effort to update and expand our MRI–immune–genetic approach in a more actual contextualization.

## 5. Conclusions

The present study suggests that interlacing MR Imaging and genetic features with tissue immune characteristics might provide suitable risk stratification models to dissect GB clinical outcome, potentially offering new therapeutic targets.

## Figures and Tables

**Figure 1 cancers-14-03249-f001:**
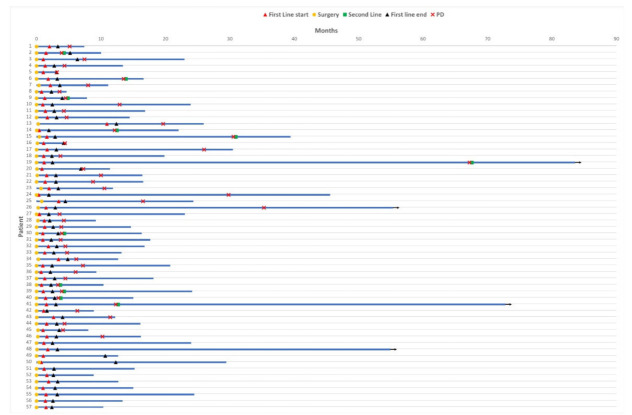
Patient clinical course. Swimmer plot to illustrate patient clinical course, including the main oncological events: surgery (yellow circle), first line treatment (start—red triangle; end—black triangle), disease progression (PD, red cross) and eventual second line treatment (green square). At data cut-off, 3 patients were still alive (black arrow). To be noted: MRI date are not plotted because essentially overlapping with surgery, as MRI was performed within 5 days prior to surgery.

**Figure 2 cancers-14-03249-f002:**
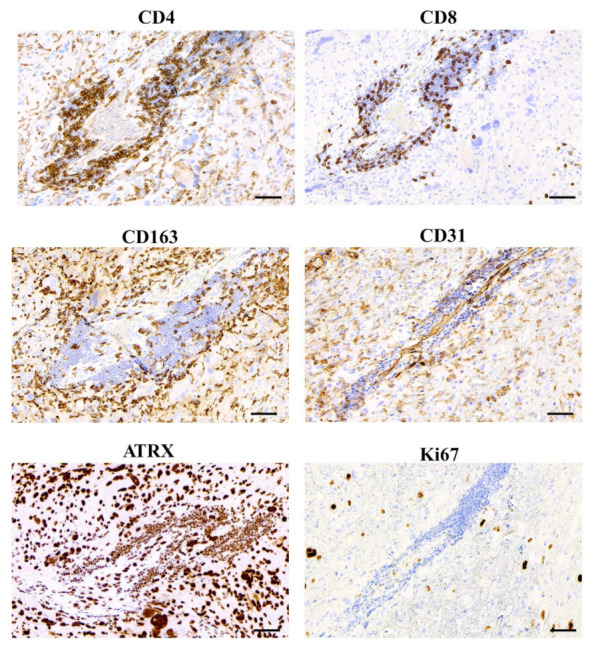
Glioblastoma Immune Microenvironment. Immunoperoxidase stained serial sections of a surgically resected glioblastoma to illustrate on the same microscopic field CD4+ and CD8+ tumor infiltrating lymphocytes (TILs) and CD163+ tumor associated macrophages (TAMs). The vascular profile, recognized by CD31+ (PECAM-1) endothelial cells lining the lumen filled by unstained red blood cells, is predominantly surrounded and infiltrated by CD4+ TILs. The remarkable contribution of CD163+ TAMs to the perivascular immune microenvironment is apparent. Nuclear ATRX staining of angio-invasive cancer cells of different dimensions, including giant cells, is shown in a serial section in which the vascular structure is barely recognized by small ATRX positive intravascular lymphocytes and few stromal-vascular ATRX negative cells (**bottom left**). The proliferative boost of glioblastoma cells is depicted by nuclear Ki67 labelling. Nuclear blue counterstaining by light Hematoxylin. Scale bars: 50 µm.

**Figure 3 cancers-14-03249-f003:**
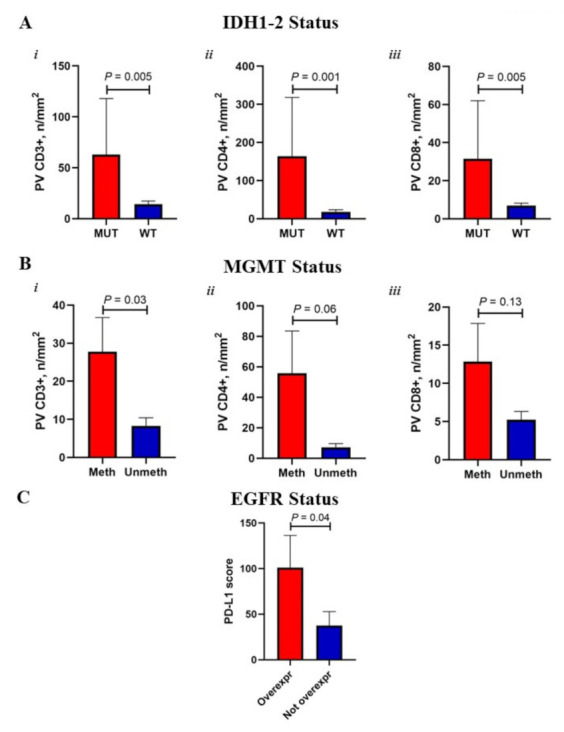
Correlations between TIME and genetic-molecular characteristics. Bar charts reporting Mean (+St.Err) values of perivascular CD3+ (***i***), CD4+ (***ii***) and CD8+ (***iii***) TILs according to IDH1-2 (**A**) mutational and MGMT (**B**) methylation status. (**C**): bar graphs illustrating the extent of PD-L1 tumor score according to EGFR expression.

**Figure 4 cancers-14-03249-f004:**
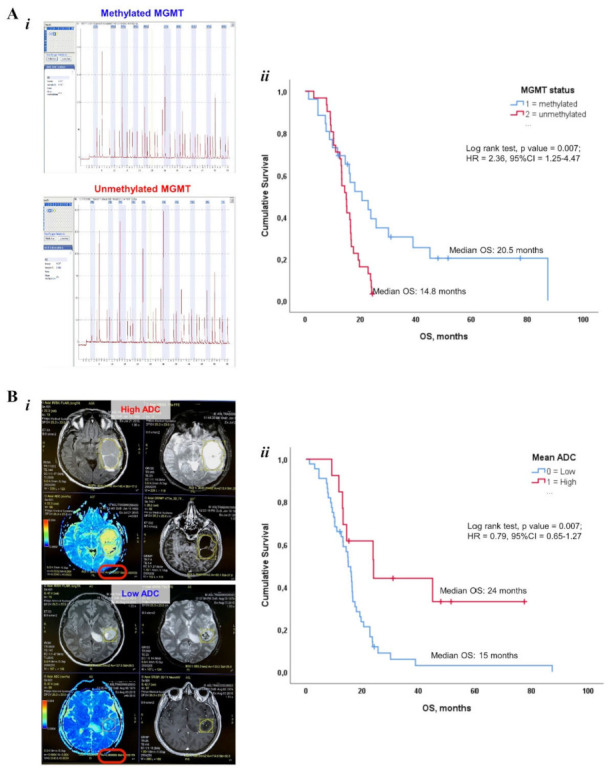
Impact of genetic and neuroradiologic features on survival outcome. (**A*i***): Representative MGMT pyrosequencing of methylated (upper) and unmethylated (lower) GB samples. (**A*ii***): Kaplan–Meier survival curves showing the impact of MGMT methylation status on OS. (**B*i***): Representative MRI illustrating high (upper) and low (lower) apparent diffusion coefficient (ADC) values, which are encircled in red. (**B*ii***): Kaplan–Meier survival curves documenting the impact of MRI-based Mean ADC on OS.

**Figure 5 cancers-14-03249-f005:**
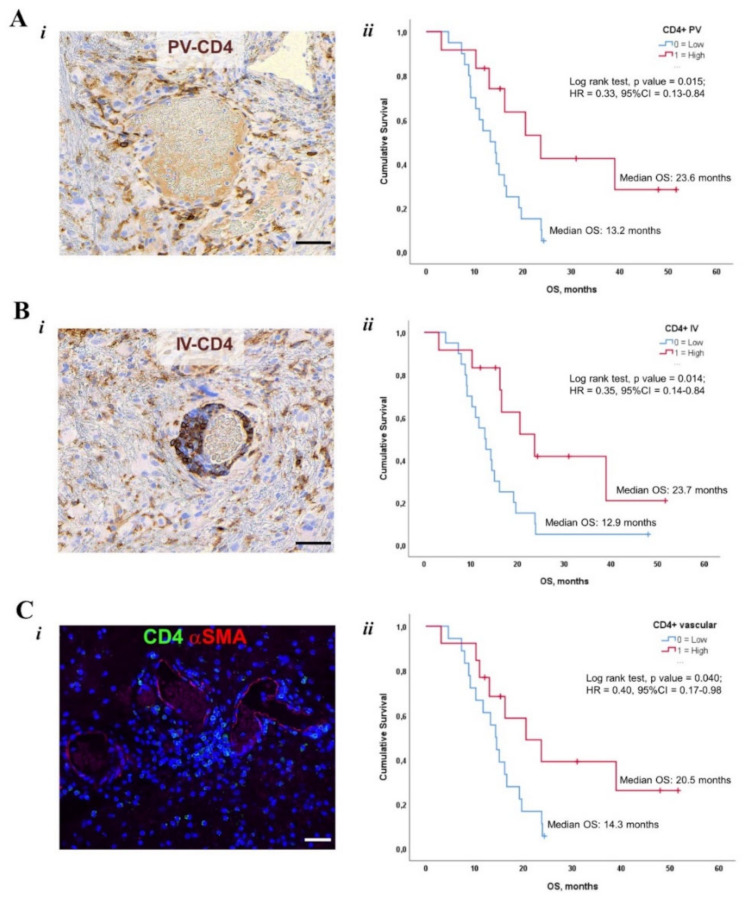
Impact of TIME features on survival outcome. (**A*i***–**B*i***): Representative images of immunoperoxidase stained sections from glioblastoma samples documenting the perivascular (PV, (**A*i***)) and intravascular (IV, (**B*i***)) localization of CD4+ (brownish) lymphocytes. (**C*i***): double immunofluorescence staining on a section of glioblastoma to simultaneously detect CD4 (green) and a-Smooth Muscle Actin (a-SMA, red). CD4+ lymphocytes are aggregated in a PV cluster or in contact with a-SMA cells. Scale Bars: 50µm. Kaplan–Meier survival curves documenting Overall Survival (OS) according to the incidence (*n*/mm^2^) of PV (**A*ii***), IV (**B*ii***) and overall vascular (PV+IV, (**C*ii***)) CD4+ TILs, respectively.

**Figure 6 cancers-14-03249-f006:**
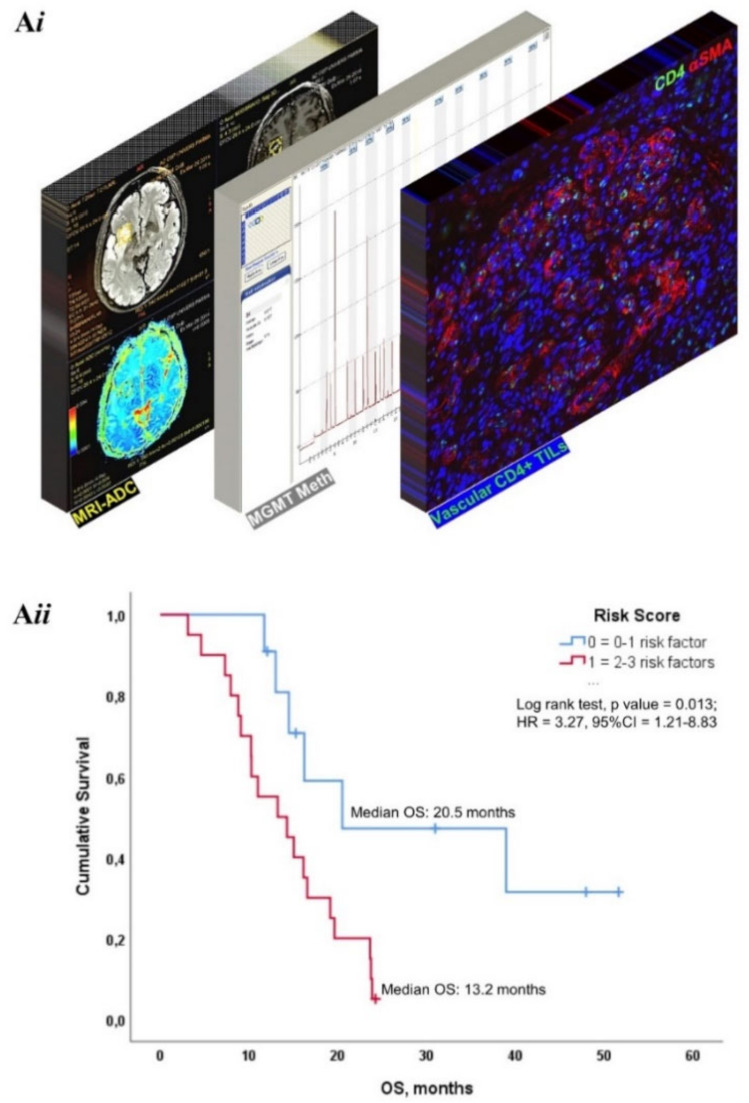
Highly Prognostic Integrated Risk Stratification Model. (**A*i***): Schematic representation of our approach to generate a prognostic score interlacing MRI-derived ADC value, MGMT methylation status and the incidence of vascular CD4+ lymphocytes, as pre-determined risk factors. Representative images of original data obtained from our patient cohort and adapted for illustrative purposes. (**A*ii***): Kaplan–Meier curve documenting patient survival (OS) according to the presence and extent of low mean ADC values, MGMT not methylated and low vascular CD4+ TILs.

**Table 1 cancers-14-03249-t001:** Patient population.

	Total (*n* = 57)	
**Age, years (Median, range)**		63 (41–82)
**Overall Survival (OS, median, range)**		15 (1–87)
		***n* (%)**
**Sex**	**Male**	33 (58)
	**Female**	24 (42)
**ECOG PS**	**0–1**	53 (93)
	**2**	4 (7)
**Site of primary lesions**	**Frontal lobe**	26 (45)
	**Temporal lobe**	50 (35)
	**Parietal**	5 (9)
	**Occipital**	4 (7)
	**Cerebellar**	1 (2)
	**Deep**	1 (2)
**Number of lesions at diagnosis**	**Single**	45 (79)
	**Multiple**	12 (21)
**Genetic-molecular status**		
**IDH1-2 ***	**WT**	53 (88)
	**Mutant**	3 (5)
**EGFR**	**Not overexpressed**	32 (56)
	**Overexpressed**	25 (44)
**p53**	**WT**	22 (44)
	**Mutant**	35 (56)
**MGMT**	**Not methylated**	31 (54)
	**Methylated**	26 (46)
**First-line treatment**	**STUPP protocol**	49 (86)
	**Other protocols (RT alone; RT → CT)**	8 (14)
**Second-line treatment**	**Yes**	12 (21)
	**No**	45 (79)

ECOG: Eastern Cooperative Group Performance Status; IDH: Isocitrate dehydrogenase; EGFR: Epidermal Growth Factor Receptor; MGMT: O6-methylguanine-DNA methyltransferase; RT: Radiotherapy; CT: Chemotherapy. * In one patient the data was not available.

**Table 2 cancers-14-03249-t002:** MRI Texture Analysis.

MRI Systems		*n* (%)
- 1.5 T		21 (37)
- 3 T		36 (63)
	**Median**	**Range**
**Max area of tumor enhancement, mm^2^**	1183.95	229.7–3030.20
**Mean ADC, mm^2^/s**	1.2 × 10^−3^	0.65–3.43 × 10^−3^
**SD ADC, mm^2^/s**	2.95 × 10^−4^	0.32–9.15 × 10^−4^
**SD FLAIR, *n***	111.80	12.20–738.10

MRI: Magnetic Resonance Imaging; ADC: Apparent Diffusion Coefficient; SD: Standard Deviation; FLAIR: Fluid-attenuated inversion recovery.

**Table 3 cancers-14-03249-t003:** TIME Characteristics.

Median		Range
**CD3+ TILs, *n*/mm^2^**		
Total	37.45	9.46–360.49
IT	18.04	1.28–326.03
PV	4.88	0.29–118.65
IV	1.26	0–15.91
**CD4+ TILs,** ***n*/mm^2^**		
Total	17.35	1.63–481.16
IT	10.05	0.53–301.58
PV	2.88	0–318.02
IV	0.30	0–69.6
**CD8+ TILs, *n*/mm^2^**		
Total	16.38	3.45–265.91
IT	8.02	0.88–230.33
PV	2.64	0–62.11
IV	0.53	0–11.05
**CD163 area, %**	1.86	0.03–6.15
**PD-L1 tumor score**	12	0–270

TILs: Tumor Infiltrating Lymphocytes; IT: intratumor; PV: perivascular; IV: intravascular.

**Table 4 cancers-14-03249-t004:** Explanatory prognostic factors in Cox proportional hazard models.

OS, Univariate Analysis ^a^	Overall			
	HR	CI (95%)	χ^2^	*p* Value
**Age**	1.033	1.002–1.065	4.435	0.065
**Sex**	1.168	0.658–2.073	0.282	0.595
**Location of primary lesions**	1.156	0.923–1.447	1.600	0.206
**Number of lesions at diagnosis**	1.364	0.710–2.621	0.868	0.351
**IDH1-2**	6.179	0.841–45.409	3.203	0.074
**EGFR**	1.230	0.698–2.167	0.513	0.474
**p53**	0.757	0.422–1.359	0.871	0.351
**MGMT**	2.363	1.249–4.470	6.995	**0.008**
**CD3+ TILs, *n*/mm^2^**				
**Total**	0.995	0.988–1.002	2.003	0.157
**IT**	0.998	0.992–1.004	0.430	0.512
**PV**	0.968	0.935–1.001	3.645	0.056
**IV**	0.895	0.804–0.996	4.131	**0.042**
**CD4+ TILs, *n*/mm^2^**				
**Total**	0.994	0.988–1.000	3.582	0.058
**IT**	0.994	0.986–1.003	1.658	0.198
**PV**	0.806	0.746–1.007	2.317	**0.048**
**IV**	0.810	0.759–1.015	2.848	**0.042**
**CD8+ TILs, *n*/mm^2^**				
**Total**	0.996	0.988–1.003	1.233	0.267
**IT**	0.998	0.991–1.005	0.354	0.552
**PV**	0.949	0.899–1.002	3.598	0.078
**IV**	0.958	0.895–0.977	4.838	0.128
**CD163 area, %**	0.966	0.761–1.228	0.078	0.780
**PD-L1 tumor score**	0.997	0.992–1.002	1.297	0.255
**Max Area of Tumor Enhancement, mm^2^**	1	0.999–1.000	1.283	0.257
**Mean ADC, mm^2^/s**	0.688	0.598–1.000	8.741	**0.003**
**SD ADC, mm^2^/s**	0.998	0.996–0.999	9.131	0.113
**SD FLAIR**	0.999	0.996–1.001	1.294	0.255

OS: Overall Survival; Age (continue variable), Sex (Male = 0, Female = 1), Location of primary lesions (Frontal = 1, Temporal = 2, Parietal = 3, Occipital = 4, Cerebellar = 5, Deep = 6), Number of lesions at diagnosis (continue variable), IDH 1-2 (Mutant = 1, WT = 2), EGFR (Overexpressed = 1, Not overexpressed = 2), p53 (Mutant = 1, WT = 2), MGMT (Methylated = 1, Unmethylated = 2), CD3+/CD4+/CD8+ TILs (continue variables), CD163 area (continue variable), PD-L1 (continue variable), Max area of tumor enhancement/Mean ADC/SD ADC/SD FLAIR (continue variables). Statistical results with *p* < 0.05 are bolded. ^a^ Univariate analysis carried out without any adjustment.

## Data Availability

The data presented in this study are available on request from the corresponding author.

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
