# Peer review of "Integrated MRI–Immune–Genomic Features Enclose a Risk Stratification Model in Patients Affected by Glioblastoma"

_cancers, 2022, doi:10.3390/cancers14133249_

Round 1

Reviewer 1 Report

Study is well-designed and has strong clinical implication. Given diffuse glioma is a multi-mutational tumor, it is important to move from single variant analysis and consider the larger impact of the disease with a multi multiparameter approach to improve prognostic evaluation. Overall, no major concern. Minor comments for clarification.

1) Details on the quantification of CD3, CD8 CD4 in the material and methods section can be further improved as explanation is vague.

2) Author should include patient survival data in the table 1 summary.

3) Author should include first-line therapy (radiotherapy and TMZ) or additional treatment information in table 1 summary.

4) Is there any clinical information on 1p19q co-deletion?

5) Are perivascular regions found predominately in the tumor core or peripheral? Is there a correlation to patient survival if CD4+ staining is found differently at the tumor core or peripheral region?

6) Do the authors see a similar outcome in their prognostic value if they sperate long-term and short-term patient cohort and analyze them independently?

Reviewer 2 Report

Mazzaschi and colleagues have been working on building a multimodal model to improve prognostication of GBM patients. Using a combination of imaging, genetic profiling and tissue interrogation, the authors put forward a novel integrated model to predict patient outcome in GBM. 

I  do have some reservations regarding this study  for which the authors need to provide a more detailed description of their  cohort.

1. The authors have mixed IDH mut and wt tumors together, while in the novel WHO classification these are now considered as very different entities. It  is of course interesting to compare both groups. Considering the much better prognosis of the IDH mutant tumors, how do these patients skew or modify  the survival analysis that was performed? Ideally, all survival curves are also represented with and without the IDH mutant group. Also, regarding the EOR and treatment (eg number of cycles of STUPP in each patient) that was given to the patient, all details should be given and stratification and correlation analysis between those features should be provided. For instance, swimmer plots can be provided to show the overview of the clinical data where important events are highlighted. Also when imaging, etc was performed is key  to properly   interpret the present data. Finally, also the KPS seems to be different between the 2 groups - median of 70 vs 90. Please provide a jitter plot to see the distributions better.

2. There is a clear correlation of MGMT methylation status with survival (again please redo these analysis without the IDH mutant patients) as has been reported extensively before. In this study, the authors also find that the CD4 PV features seems to correlate strongly to MGMT methylation status. Considering this observation, it is not surprising that CD4 PV will then also correlate to a better survival, even though this may just be a bystander effect. The authors should try to study whether both features are biologically  linked or whether these are independent events co-occuring. At least a proper discussion about this observation should be made throughout the manuscript/discussion. Connected to this, while the presence of CD4 T cells is an interesting parameter, do the authors have any  additional information regarding their status (activated, exhausted, helper subtypes, PD1+/-, proliferating, etc)?

3. The IHC analysis was done by staining serial sections using standard DAB/IHC methods. However, CD4 is also often expressed strongly in macrophages, which makes it often difficult to discern between both cell  types, certainly in PV regions; there are nowadays many tools to perform multiplexed iHC where all the various cell types can be studied in the same slide, including various T cell subtypes, Blood vessels, macrophages and tumor cells. There are also no details on the areas that were investigated, where the areas in the original tumors were selected (just random?), and which analysis tools were used to generate the data. 

4. Regarding the integration of MRI vs IHC - do the authors know the exact location in the MRI scan where tissue was collected for IHC? Are there any direct correlations that can be made?

5. Were data corrected for multiple testing? Certainly when reviewing the various modalities for correlation this is very important.
